# An Efficient Local Formulation for Time–Dependent PDEs

**Imtiaz Ahmad** [1], **Muhammad Ahsan** [1,2], **Zaheer-ud Din** [2,3] **and Masood Ahmad** [2] **and Poom Kumam** [4,5,*]

1   Department of Mathematics, University of Swabi, Swabi 23430, Pakistan; imtiazkakakhil@gmail.com (I.A.); ahsankog@uoswabi.edu.pk (M.A.)
2   Department of Basic Sciences, University of Engineering and Technology, Peshawar 25000, Pakistan; zaheer.mth@gmail.com (Z.-u.D.); masood.suf@gmail.com (M.A.)
3   Department of Basic Sciences, CECOS University of IT and Emerging Sciences, Peshawar 25000, Pakistan
4   KMUTT-Fixed Point Research Laboratory, Department of Mathematics, Room SCL 802 Fixed Point Laboratory, Science Laboratory Building, Faculty of Science, King Mongkut's University of Technology Thonburi (KMUTT), 126 Pracha-Uthit Road, Bang Mod, Thrung Khru, Bangkok 10140, Thailand
5   KMUTT-Fixed Point Theory and Applications Research Group, Theoretical and Computational Science Center (TaCS), Science Laboratory Building, Faculty of Science, King Mongkut's University of Technology Thonburi (KMUTT), 126 Pracha-Uthit Road, Bang Mod, Thrung Khru, Bangkok 10140, Thailand
*   Correspondence: poom.kum@kmutt.ac.th

**Abstract:** In this paper, a local meshless method (LMM) based on radial basis functions (RBFs) is utilized for the numerical solution of various types of PDEs. This local approach has flexibility with respect to geometry along with high order of convergence rate. In case of global meshless methods, the two major deficiencies are the computational cost and the optimum value of shape parameter. Therefore, research is currently focused towards localized RBFs approximations, as proposed here. The proposed local meshless procedure is used for spatial discretization, whereas for temporal discretization, different time integrators are employed. The proposed local meshless method is testified in terms of efficiency, accuracy and ease of implementation on regular and irregular domains.

**Keywords:** local meshless method; RBFs; irregular domains; Kortewege-de Vries types equations; reaction-diffusion Brusselator system

## 1. Introduction

Partial differential equation (PDE) of reaction-convection-diffusion type are physically very rich. These PDEs can be abundantly found in modeling physical sciences, biological sciences and mathematics of finance. Some of these models include the fifth order Kortewege-de Vries model. Its general exact solution is not known whereas the exact solution for particular case of solitary waves can be found in [1]. To solve this model numerically, serval methods can be found in literature such as finite-deference scheme [2], modified ADM [3], decomposition method [4], Homotopy perturbation transform method [5] and a comparative study of Crank-Nicolson method and ADM [6].

The general seventh order Kortewege-de Vries equation [7] which is used to discuss structural stability under singular perturbation of standard KdV equation. Several authors have paid attention to solve the seventh order KdV equation [8–10].

Generalized Burgers' Huxley equation [11] is used to describe the interaction between convection effects, reaction mechanisms and diffusion transports. In general it is difficult and sometimes impossible to get exact solution of such type of nonlinear PDEs. Researchers employed different numerical techniques which include discrete Adomian decomposition method [12], Haar wavelet

method [13], Adomian decomposition method [14], meshless collocation method based on RBF [15] and local meshless method [16] for the solution of Burgers' Huxley equation. The Huxley model equation [17] describes nerve pulse propagation in nerve fibres and wall motion in liquid crystals [18]. Numerical solution of Huxley equation can be found in [14,15,19] and the references therein. Generalized Burgers' Fisher equation [20], describe the propagation of a mutant gene. Different numerical methods have been used for numerical solution of this model, such as meshless collocation method [15], ADM [14], VIM [21], modified pseudo spectral method [22] and modified cubic B–spline functions collocation method [23].

The Fitzhugh-Nagumo (FN) equation has numerous applications in different fields such as branching brownian motion process, flame propagation, neurophysiology, logistic population growth and nuclear reactor theory [24]. Numerical solution of FN equation can be found in [23,25–27].

Hirota-Satsuma introduced the nonlinear coupled Kortewege-de Vries equations [28]. This model has numerous applications in physical sciences. In the last decade, researchers have used various numerical techniques for the solution of this model equations. These include RBFs collocation (Kansa) method [29], meshless RBFs method of lines [30], variational iteration method [31] and spectral collocation method [32]. The Hirota–Satsuma coupled KdV system [33] has been solved by different numerical methods given in [32,34,35] and the references therein.

The Brusselator system is one of the essential reaction–diffusion model equation. This model explain the mechanism of a chemical reaction–diffusion with non–linear oscillations [36,37]. Numerical solution of this kind of model can be found in [38–41].

The horizon of meshless methods is continuously expanding and their applicability is increasing rapidly due to their ease of implementation in higher dimensions on a set of uniform or scattered data points in regular and irregular geometries. In the last few years, it is observed that meshless methods have been extensively employed for numerical simulations of different types of PDEs [29,42–44]. Meshless methods reduce complexity caused due to dimensionality to a large extent which is being faced in the carrying out of conventional methods like finite-element and finite-difference procedures. Meshing in the case of complicated geometries is another cause for the growing demand of meshless methods.

It is noticed that the global meshless method (GMM) which is based on global interpolation paradigm faced the problem of dense ill-conditioned matrices and finding optimum value of the shape parameter. To avoid limitations of the GMM, a local meshless method which is based on local interpolation in the sub-domains are used as substitute to get a stable and accurate solution for the PDE models [43,45–48].

Organization of rest of the paper is as follows; In Section 2, we describe the models briefly. In Section 3, we highlight the proposed method. In Sections 4, the numerical methods are applied to different test problems and the results are compared with published work. In Section 5, some conclusions are drawn.

## 2. Partial Differential Equation Models

A short description of PDE models on a bounded domain with corresponding initial and boundary conditions are given in this section. These include; one-dimensional fifth order Lax's-Kortewege-de Vries, seventh order Lax's-Kortewege-de Vries, generalized Burgers'-Huxley, Huxley, generalized Burgers' Fisher, Fitzhugh-Nagumo, coupled Kortewege-de Vries, Hirota-Satsuma coupled Kortewege-de Vries equations and two-dimensional reaction-diffusion Brusselator equations.

The 1D fifth order Kortewege-de Vries equation [3],

$$U_t + aU^2 U_x + bU_x U_{xx} + cUU_{xxx} + dU_{xxxxx} = 0, \tag{1}$$

where $a, b, c, d$ are real constants and in this paper we have taken $a = b = 30, c = 10$, and $d = 1$.

The 1D seventh order Kortewege-de Vries equation [9],

$$U_t + aU^3U_x + bU_x^3 + cUU_xU_{xx} + dU^2U_{xxx} + eU_{xx}U_{xxx} + fU_xU_{xxxx} + gUU_{xxxxx} + U_{xxxxxxx} = 0, \quad (2)$$

where $a, b, c, d, e, f, g$ are real constants and we have taken $a = 140$, $b = 70$, $c = 280$, $d = 70$, $e = 70$, $f = 42$, $g = 14$.

The 1D generalized Burgers'-Huxley equation [11–15],

$$U_t + \alpha U^\delta U_x - U_{xx} - \beta U(1 - U^\delta)(U^\delta - \gamma) = 0, \quad (3)$$

where $\alpha, \beta \geq 0$, $\delta > 0$, $\gamma \in (0, 1)$ are constants.

The 1D Huxley equation [14,15,19],

$$U_t - U_{xx} - \beta U(1 - U)(U - \gamma) = 0, \quad (4)$$

where $\beta$ and $\gamma$ are constants.

The 1D generalized Burgers' Fisher equation [14,15],

$$U_t + \alpha U^\delta U_x - U_{xx} - \beta U(1 - U^\delta) = 0, \quad (5)$$

where $\alpha, \beta$ and $\delta$ are constants.

The 1D Fitzhugh-Nagumo (FN) equation [24,27,49,50],

$$U_t - U_{xx} + U(1 - U)(\rho - U) = 0, \quad (6)$$

where $\rho$ is a constant.

The 1D coupled Kortewege-de Vries equation [28–30,32],

$$\begin{aligned} U_t + 6\alpha UU_x - 2\gamma VV_x + \alpha U_{xxx} &= 0, \\ V_t + 3\beta UV_x + \beta V_{xxx} &= 0, \end{aligned} \quad (7)$$

where $\alpha, \beta$ and $\gamma$ are real parameters.

The 1D Hirota-Satsuma coupled KdV system of equations [32,33],

$$\begin{aligned} U_t - \frac{1}{2}U_{xxx} + 3UU_x - 3VW_x - 3WV_x &= 0, \\ V_t + V_{xxx} - 3UV_x &= 0, \\ W_t + W_{xxx} - 3UW_x &= 0. \end{aligned} \quad (8)$$

The 2D reaction-diffusion Brusselator system of equations [38,41],

$$\begin{aligned} U_t - \beta - U^2V + (\alpha + 1)U - \gamma(U_{xx} + U_{yy}) &= 0, \\ V_t - \alpha U + U^2V - \gamma(V_{xx} + V_{yy}) &= 0, \end{aligned} \quad (9)$$

where $\alpha, \beta$ and $\gamma$ are constants.

## 3. Local Meshless Numerical Scheme

To pursue the LMM [43,51], we approximate the derivatives of $U(x, t)$ at the center $x_p$ by the function values at a set of nodes in the neighborhood of $x_p$, $\{x_{p_1}, x_{p_2}, x_{p_3}, ..., x_{p_{n_j}}\} \subset \{x_1, x_2, \ldots, x_{N^n}\}$, $n_p \ll N^n$, where $n = 1$, $n = 2$ for one and two dimensional case respectively.

$$U^{(m)}(x_p) \approx \sum_{k=1}^{n_p} \lambda_k^{(m)} U(x_{p_k}), \; p = 1, 2, \ldots, N^n. \quad (10)$$

To find the corresponding coefficient $\lambda_k^{(m)}$, radial basis function $\phi(\|x - x_l\|)$ can be substituted into Equation (10)

$$\phi^{(m)}(\|x_p - x_l\|) = \sum_{k=1}^{n_p} \lambda_{p_k}^{(m)} \phi(\|x_{p_k} - x_l\|), \; l = p_1, p_2, \ldots, p_{n_p}. \tag{11}$$

Equation (11) in matrix form

$$\begin{bmatrix} \phi_{p_1}^{(m)}(x_p) \\ \phi_{p_2}^{(m)}(x_p) \\ \vdots \\ \phi_{p_{n_p}}^{(m)}(x_p) \end{bmatrix} = \begin{bmatrix} \phi_{p_1}(x_{p_1}) & \phi_{p_2}(x_{p_1}) & \cdots & \phi_{p_{n_p}}(x_{p_1}) \\ \phi_{p_1}(x_{p_2}) & \phi_{p_2}(x_{p_2}) & \cdots & \phi_{p_{n_p}}(x_{p_2}) \\ \vdots & \vdots & \ddots & \vdots \\ \phi_{p_1}(x_{p_{n_p}}) & \phi_{p_2}(x_{p_{n_p}}) & \cdots & \phi_{p_{n_p}}(x_{p_{n_p}}) \end{bmatrix} \begin{bmatrix} \lambda_{p_1}^{(m)} \\ \lambda_{p_2}^{(m)} \\ \vdots \\ \lambda_{p_{n_p}}^{(m)} \end{bmatrix}, \tag{12}$$

where

$$\phi_l(x_k) = \phi(\|x_k - x_l\|), \; l = p_1, p_2, \ldots, p_{n_p}, \tag{13}$$

for each $k = p_1, p_2, \ldots, p_{n_p}$.

The above equation in matrix notation

$$\mathbf{\Phi}_{n_p}^{(m)} = \mathbf{A}_{n_p} \lambda_{n_p}^{(m)}, \tag{14}$$

where

$$\mathbf{\Phi}_{n_p}^{(m)} = \begin{bmatrix} \phi_{p_1}^{(m)}(x_p) & \phi_{p_2}^{(m)}(x_p) & \cdots & \phi_{p_{n_p}}^{(m)}(x_p) \end{bmatrix}^T$$

$$\mathbf{A}_{n_p} = \begin{bmatrix} \phi_{p_1}(x_{p_1}) & \phi_{p_2}(x_{p_1}) & \cdots & \phi_{p_{n_p}}(x_{p_1}) \\ \phi_{p_1}(x_{p_2}) & \phi_{p_2}(x_{p_2}) & \cdots & \phi_{p_{n_p}}(x_{p_2}) \\ \vdots & \vdots & \ddots & \vdots \\ \phi_{p_1}(x_{p_{n_p}}) & \phi_{p_2}(x_{p_{n_p}}) & \cdots & \phi_{p_{n_p}}(x_{p_{n_p}}) \end{bmatrix}$$

$$\lambda_{n_p}^{(m)} = \begin{bmatrix} \lambda_{p_1}^{(m)} & \lambda_{p_2}^{(m)} & \cdots & \lambda_{p_{n_p}}^{(m)} \end{bmatrix}^T.$$

From Equation (14),

$$\lambda_{n_p}^{(m)} = \mathbf{A}_{n_p}^{-1} \mathbf{\Phi}_{n_p}^{(m)}. \tag{15}$$

By substituting Equation (15) into Equation (10),

$$U^{(m)}(x_p) = (\lambda_{n_p}^{(m)})^T \mathbf{U}_{n_p}, \tag{16}$$

where

$$\mathbf{U}_{n_p} = \begin{bmatrix} U(x_{p_1}), U(x_{p_2}), \ldots, U(x_{p_{n_p}}) \end{bmatrix}^T. \tag{17}$$

### 3.1. 1D Fifth Order Kortewege-de Vries Equation

The 1D fifth order Kortewege-de Vries Equation (1) can be written as

$$U_t = -(aU^2 U_x + bU_x U_{xx} + cUU_{xxx} + dU_{xxxxx}), \; x \in \Omega, \; t \geq 0, \tag{18}$$

subject to initial and boundary conditions

$$U(x, 0) = f(x), \; x \in \Omega, \tag{19}$$

$$U(x,t) = f_1(t), \quad U(x,t) = f_2(t), \quad x \in \partial\Omega. \tag{20}$$

where $a$, $b$, $c$ and $d$ are constants.

Now, Applying the LMM to Equation (18) we get,

$$\frac{dU_p}{dt} = -(aU_p^2(\lambda_{n_p}^{(1)})^T \mathbf{U}_{n_p} + b((\lambda_{n_p}^{(1)})^T \mathbf{U}_{n_p})((\lambda_{n_p}^{(2)})^T \mathbf{U}_{n_p}) + cU_p(\lambda_{n_p}^{(3)})^T \mathbf{U}_{n_p} + d(\lambda_{n_p}^{(5)})^T \mathbf{U}_{n_p}), \quad p = 2,3,\ldots,N-1. \tag{21}$$

The semi-discretized model Equation (21) with boundary conditions (20) is given as follows

$$\frac{d\mathfrak{U}}{dt} = -(a\,\mathfrak{U}^2 * (\mathbf{\Lambda}^{(1)}\mathfrak{U}) + b(\mathbf{\Lambda}^{(1)}\mathfrak{U}) * (\mathbf{\Lambda}^{(2)}\mathfrak{U}) + c\,\mathfrak{U} * (\mathbf{\Lambda}^{(3)}\mathfrak{U}) + d(\mathbf{\Lambda}^{(5)}\mathfrak{U})), \tag{22}$$

where the symbol $*$ represent element-wise multiplication of two vectors.

$$\begin{aligned}
\mathfrak{U} &= [U_1, U_2, U_3, \ldots, U_N]^T, \\
\mathbf{\Lambda}_{N \times N}^{(1)} &= [m_{pk}] = [\lambda_k^{(1)}], k = p_1, p_2, \ldots, p_{n_p}, p = 2,3,\ldots,N-1, \\
\mathbf{\Lambda}_{N \times N}^{(2)} &= [m_{pk}] = [\lambda_k^{(2)}], k = p_1, p_2, \ldots, p_{n_p}, p = 2,3,\ldots,N-1. \\
\mathbf{\Lambda}_{N \times N}^{(3)} &= [m_{pk}] = [\lambda_k^{(3)}], k = p_1, p_2, \ldots, p_{n_p}, p = 2,3,\ldots,N-1. \\
\mathbf{\Lambda}_{N \times N}^{(5)} &= [m_{pk}] = [\lambda_k^{(5)}], k = p_1, p_2, \ldots, p_{n_p}, p = 2,3,\ldots,N-1.
\end{aligned} \tag{23}$$

The corresponding initial condition is given as

$$\mathfrak{U}(t_0) = [U_0(x_1), U_0(x_2), \ldots, U_0(x_N)]^T. \tag{24}$$

## 4. Results and Discussion

To check the accuracy and efficiency of the LMM various test problems in one and two dimensional cases are considered and the results are compared with the existence methods reported in literature. For spatial discretization three types of RBFs that is, multiquadric (MQ), inverse multiquadric (IMQ) and Gaussian (GA) are used whereas for time integration have used explicit Euler method (EEM) and Runge-Kutta method of order 4 (RK4).

Accuracy of the LMM is measured though different error norms given as follows

$$\begin{aligned}
L_{abs} &= |U_{exact}(p) - U(p)|, \quad p = 1,2,\ldots,N^n. \\
L_\infty &= \max\left(|U_{exact}(p) - U(p)|\right), \\
L_2 &= \left[\Delta x \sum_{p=1}^{N} L_{abs}^2\right]^{\frac{1}{2}}, \\
L_{rms} &= \left[\frac{1}{N} \sum_{p=1}^{N} L_{abs}^2\right]^{\frac{1}{2}},
\end{aligned} \tag{25}$$

where exact and numerical solution are represented by $U_{exact}$ and $U$ respectively.

In this paper, we have considered both uniform and non-uniform nodal points. In 1D case, the size of local sub domain is taken $n_p = 3$ whereas in 2D case the size of the local sub domain is taken $n_p = 5$ for all the numerical experiments. The Central Processing Unit (CPU) time is calculated in seconds in all the cases.

Summary of numerical results is given as: Results of 1D fifth order KdV equation are shown in Table 1 and compared with the method in [3] whereas numerical results of seventh order KdV equation are presented in Table 2. Similarly the numerical results of generalized Burgers' Huxley equation are presented in Tables 3 and 4 and the results are compared with the methods given in [12–15] while the numerical results of Huxley equation are shown in Tables 5 and 6 and compared with the methods

in [14,15,19]. Numerical results of generalized Burgers' Fisher equation are shown in Table 7 and compared with the methods in [14,15]. Numerical results of Fitzhugh-Nagumo equation are given in Table 8, Figures 1 and 2 and the results are compared with the method reported in [27]. Numerical results of coupled KdV equations are presented in Table 9 and the results are compared with the method given in [29] while numerical results of Hirota-Satsuma coupled KdV equation are shown in Table 10. Numerical simulation of reaction-diffusion Brusselator system are shown in Table 11 and comparison is made with the methods in [38,41]. All the computations were performed on Dell PC Laptop with an Intel(R) Core(TM)i5-2450M CPU 2.50GHz 2.50GHz 4 GB RAM.

**Problem 1.** *The exact solution [3] of the 1D Lax's fifth order KdV Equation* (1) *is*

$$U(x,t) = 2k^2 \left( 2 - 3\tanh^2 \left( k(x - 56k^4 t - x_0) \right) \right), \quad x \in [-10, 10], \quad t \geq 0 \tag{26}$$

*where the initial and boundary conditions are extracted from the exact solution* (26).

Numerical results for Test Problem 1 are given in Table 1 using $k = 0.01$, $x_0 = 0$, $\Delta t = 0.01$, $N = 11$ and shape parameter $c = 100$. Table 1 indicated that the results produced by the LMM using EEM are more better than the method in [3].

**Table 1.** Comparisons between the results obtained by the LMM with those adapted from [3] for Test Problem 1.

| | x = 0.2 | | x = 1 | | x = 5 | |
|---|---|---|---|---|---|---|
| *t* | **EEM** | **[3]** | **EEM** | **[3]** | **EEM** | **[3]** |
| 2 | $5.2899 \times 10^{-15}$ | $5.7619 \times 10^{-14}$ | $2.6450 \times 10^{-14}$ | $2.8786 \times 10^{-13}$ | $1.2669 \times 10^{-13}$ | $1.4210 \times 10^{-12}$ |
| 4 | $1.0580 \times 10^{-14}$ | $1.1528 \times 10^{-13}$ | $5.2900 \times 10^{-14}$ | $5.7577 \times 10^{-13}$ | $2.5337 \times 10^{-13}$ | $2.8421 \times 10^{-12}$ |
| 6 | $1.5869 \times 10^{-14}$ | $1.7298 \times 10^{-13}$ | $7.9350 \times 10^{-14}$ | $8.6372 \times 10^{-13}$ | $3.8006 \times 10^{-13}$ | $4.2632 \times 10^{-12}$ |
| 8 | $2.1159 \times 10^{-14}$ | $2.3073 \times 10^{-13}$ | $1.0580 \times 10^{-13}$ | $1.1517 \times 10^{-12}$ | $5.0676 \times 10^{-13}$ | $5.6844 \times 10^{-12}$ |
| 10 | $2.6448 \times 10^{-14}$ | $2.8851 \times 10^{-13}$ | $1.3225 \times 10^{-13}$ | $1.4397 \times 10^{-12}$ | $6.3345 \times 10^{-13}$ | $7.1056 \times 10^{-12}$ |

**Problem 2.** *The 1D Lax's seventh order KdV Equation* (2) *having exact solution [9]*

$$U(x,t) = 2k^2 \operatorname{sech}^2 \left( k(x - 64k^6 t) \right), \quad x \in [-100, 100], \quad t \geq 0 \tag{27}$$

*where the initial and boundary equations are extracted from the exact solution* (27).

To demonstrate the accuracy and efficiency of the proposed LMM, we reported numerical results in Table 2 for Test Problem 2, in form of $L_\infty$ error norm using different values of $k$ and $t$. We have used EEM with $\Delta t = 0.01$, $N = 11$ using MQ RBF ($c = 100$). From Table 2, one can observe that the LMM is accurate and efficient.

**Table 2.** Numerical results in form of $L_\infty$ error norm using the EEM for Test Problem 2.

| *t* | **k = 0.1** | **k = 0.01** | **k = 0.001** | **CPU Time** |
|---|---|---|---|---|
| 1 | $5.8157 \times 10^{-10}$ | $1.2395 \times 10^{-15}$ | $3.3881 \times 10^{-19}$ | 0.03 |
| 10 | $5.8205 \times 10^{-9}$ | $1.2395 \times 10^{-14}$ | $3.3881 \times 10^{-18}$ | 0.21 |
| 20 | $1.1652 \times 10^{-8}$ | $2.4790 \times 10^{-14}$ | $6.7763 \times 10^{-18}$ | 0.46 |
| 30 | $1.7494 \times 10^{-8}$ | $3.7185 \times 10^{-14}$ | $1.0164 \times 10^{-17}$ | 0.71 |
| 50 | $2.9211 \times 10^{-8}$ | $6.1975 \times 10^{-14}$ | $1.6941 \times 10^{-17}$ | 1.07 |

**Problem 3.** *The 1D generalized Burgers' Huxley equation* (3) *having exact solution taken from* [52] *is given by*

$$U(x,t) = \left(\frac{\gamma}{2} + \frac{\gamma}{2}\tanh\left(\omega_1(x - \omega_2 t)\right)\right)^{\frac{1}{\delta}}, \quad a \le x \le b, \ t \ge 0, \tag{28}$$

*where*

$$\omega_1 = \frac{-\alpha\delta + \delta\sqrt{\alpha^2 + 4\beta(1 + \delta)}}{4(1 + \delta)}\gamma, \quad \omega_2 = \frac{\alpha\gamma}{1 + \delta} - \frac{(1 + \delta - \gamma)(-\alpha + \sqrt{\alpha^2 + 4\beta(1 + \delta)})}{2(1 + \delta)},$$

*where $\alpha$, $\beta$, $\delta$ and $\gamma$ are constants such that $\beta \ge 0$, $\delta > 0$, $\gamma \in (0, 1)$.*
*The initial and boundary conditions are drawn out from the exact solution* (28).

In Table 3, we have compared the results obtained by the LMM for generalized Burgers' Huxley equation for Test Problem 3 with the methods given in [12,14,15]. We have used the parameters values $\alpha = \beta = \delta = 1$ and $\gamma = 0.001$ and time step length $\Delta t = 0.0001$, spatial domain $[-10, 20]$, $N = 61$ using IMQ RBF. From Table 3, we have noted that the RK4 produced more accurate results than the results reported in [12,14,15].

**Table 3.** Comparisons between the results obtained by the LMM with those adapted from [12,14,15] in term of $L_{abs}$ error norm for Test Problem 3.

| $t$ | $x$ | RK4 | [15] | [14] | [12] |
|-----|-----|-----|------|------|------|
| 0.05 | 0.1 | $6.30 \times 10^{-12}$ | $1.0 \times 10^{-9}$ | $1.93 \times 10^{-7}$ | $1.87 \times 10^{-8}$ |
|      | 0.5 | $4.42 \times 10^{-12}$ | $1.0 \times 10^{-9}$ | $1.93 \times 10^{-7}$ | $1.87 \times 10^{-8}$ |
|      | 0.9 | $2.55 \times 10^{-12}$ | $1.0 \times 10^{-9}$ | $1.93 \times 10^{-7}$ | $1.87 \times 10^{-8}$ |
| 0.1 | 0.1 | $1.23 \times 10^{-11}$ | $1.0 \times 10^{-9}$ | $3.87 \times 10^{-7}$ | $3.75 \times 10^{-8}$ |
|     | 0.5 | $8.62 \times 10^{-12}$ | $1.0 \times 10^{-9}$ | $3.87 \times 10^{-7}$ | $3.75 \times 10^{-8}$ |
|     | 0.9 | $4.87 \times 10^{-12}$ | $1.0 \times 10^{-9}$ | $3.87 \times 10^{-7}$ | $3.75 \times 10^{-8}$ |
| 1.0 | 0.1 | $8.16 \times 10^{-11}$ | $0.0 \times 10^{-9}$ | $3.88 \times 10^{-6}$ | $3.75 \times 10^{-7}$ |
|     | 0.5 | $4.41 \times 10^{-11}$ | $0.0 \times 10^{-9}$ | $3.88 \times 10^{-6}$ | $3.75 \times 10^{-7}$ |
|     | 0.9 | $6.61 \times 10^{-12}$ | $0.0 \times 10^{-9}$ | $3.88 \times 10^{-6}$ | $3.75 \times 10^{-7}$ |

Table 4 also shows the comparison of numerical results produced by the LMM with the results of Haar wavelet method given in [13]. In the table we have calculated the absolute errors for different values of $x$ and $\delta$ with $\alpha = \beta = 1$, $\gamma = 0.001$, and $t = 0.8$ using IMQ RBF. It can be observed from the table that the LMM is more accurate than the method reported in [13].

**Table 4.** Comparisons between the results obtained by the LMM with those adapted from [13] in term of $L_{abs}$ error norm for Test Problem 3.

| $x$ | $\delta = 1$ RK4 | $\delta = 1$ [13] | $\delta = 2$ RK4 | $\delta = 2$ [13] |
|-----|------|------|------|------|
| 0.15625 | $6.8575 \times 10^{-11}$ | $2.4648 \times 10^{-8}$ | $5.9052 \times 10^{-7}$ | $1.1465 \times 10^{-6}$ |
| 0.28125 | $5.9200 \times 10^{-11}$ | $3.7832 \times 10^{-8}$ | $5.8977 \times 10^{-7}$ | $1.7600 \times 10^{-6}$ |
| 0.34375 | $5.4512 \times 10^{-11}$ | $4.2226 \times 10^{-8}$ | $5.8940 \times 10^{-7}$ | $1.9644 \times 10^{-6}$ |
| 0.46875 | $4.5137 \times 10^{-11}$ | $4.6621 \times 10^{-8}$ | $5.8866 \times 10^{-7}$ | $2.1749 \times 10^{-6}$ |
| 0.53125 | $4.0448 \times 10^{-11}$ | $4.6622 \times 10^{-8}$ | $5.8828 \times 10^{-7}$ | $2.1749 \times 10^{-6}$ |
| 0.65625 | $3.1070 \times 10^{-11}$ | $4.2228 \times 10^{-8}$ | $5.8754 \times 10^{-7}$ | $1.9643 \times 10^{-6}$ |
| 0.71875 | $2.6382 \times 10^{-11}$ | $3.7834 \times 10^{-8}$ | $5.8717 \times 10^{-7}$ | $1.7603 \times 10^{-6}$ |
| 0.84375 | $1.7004 \times 10^{-11}$ | $2.4650 \times 10^{-8}$ | $5.8642 \times 10^{-7}$ | $1.1462 \times 10^{-6}$ |
| 0.96875 | $7.6260 \times 10^{-12}$ | $5.5962 \times 10^{-9}$ | $5.8568 \times 10^{-7}$ | $2.6037 \times 10^{-7}$ |

The numerical results of Huxley equation for Test Problem 3 with spatial domain $[-10, 20]$, $N = 61$, $\Delta t = 0.01$ and different values of $x$ and $t$ are shown in Table 5. We have used IMQ radial basis function and $\beta = \delta = 1$, $\gamma = 0.001$. The numerical simulations have carried out by using the RK4 and comparison is done with [14,15] in Table 5. From the table, we have noticed that the results produced by the LMM are better than the methods reported in [14,15].

**Table 5.** Comparisons between the results obtained by the LMM with those adapted from [14,15] in term of $L_{abs}$ error norm for Test Problem 3.

| $t$ | $x$ | RK4 | [15] | [14] |
|-----|-----|-----|------|------|
| | 0.1 | $2.18 \times 10^{-11}$ | $0.0 \times 10^{-9}$ | $1.88 \times 10^{-7}$ |
| 0.05 | 0.5 | $1.83 \times 10^{-11}$ | $1.0 \times 10^{-9}$ | $1.87 \times 10^{-7}$ |
| | 0.9 | $1.47 \times 10^{-11}$ | $1.0 \times 10^{-9}$ | $1.87 \times 10^{-7}$ |
| | 0.1 | $4.29 \times 10^{-11}$ | $1.0 \times 10^{-9}$ | $3.75 \times 10^{-7}$ |
| 0.1 | 0.5 | $3.59 \times 10^{-11}$ | $0.0 \times 10^{-9}$ | $3.75 \times 10^{-7}$ |
| | 0.9 | $2.88 \times 10^{-11}$ | $0.0 \times 10^{-9}$ | $3.75 \times 10^{-7}$ |
| | 0.1 | $3.18 \times 10^{-10}$ | $1.0 \times 10^{-9}$ | $3.75 \times 10^{-6}$ |
| 1.0 | 0.5 | $2.47 \times 10^{-10}$ | $0.0 \times 10^{-9}$ | $3.75 \times 10^{-6}$ |
| | 0.9 | $1.76 \times 10^{-10}$ | $1.0 \times 10^{-9}$ | $3.75 \times 10^{-6}$ |

**Problem 4.** *The exact solution [18] of the 1D Huxley Equation (4) with $\alpha = \gamma = 1$ is given below as*

$$U(x,t) = \frac{1}{2} + \frac{1}{2} \tanh\left(\frac{1}{2\sqrt{2}}(x - \frac{t}{\sqrt{2}})\right), \quad a \leq x \leq b, \quad t \geq 0. \tag{29}$$

The numerical simulations have carried out for Test Problem 4 in Table 6 for different values of $t$, $x$, $a$, $b$ and for $N = 10$, $\Delta t = 0.0001$ using MQ RBF with $c = 5$. The results are obtained by the EEM and compared with the results obtained by Chebyshev spectral collocation method in [19] and found that the results of the LMM are superior.

**Table 6.** Comparisons between the results obtained by the LMM with those adapted from [19] in term of $L_{abs}$ error norm for Test Problem 4.

| $a$ | $b$ | $t$ | $x$ | EEM | [19] |
|-----|-----|-----|-----|-----|------|
| $-2$ | 2 | 0.002 | 0.050 | $1.9974 \times 10^{-7}$ | $2.22 \times 10^{-3}$ |
| $-2$ | 2 | 0.100 | 0.700 | $3.0067 \times 10^{-6}$ | $1.78 \times 10^{-3}$ |
| $-5$ | 5 | 0.001 | 0.500 | $2.5260 \times 10^{-7}$ | $1.69 \times 10^{-2}$ |
| $-5$ | 5 | 0.001 | 2.500 | $1.7843 \times 10^{-6}$ | $9.90 \times 10^{-3}$ |
| $-10$ | 10 | 0.002 | 0.010 | $3.6673 \times 10^{-6}$ | $4.05 \times 10^{-4}$ |
| $-10$ | 10 | 0.100 | 1.000 | $2.7722 \times 10^{-4}$ | $3.17 \times 10^{-3}$ |

**Problem 5.** *The exact solution of the 1D generalized Burgers' Fisher Equation (5) is given below as*

$$U(x,t) = \left(\frac{1}{2} + \frac{1}{2} \tanh(a_1(x - a_2 t))\right)^{\frac{1}{\delta}}, \quad a \leq x \leq b, \quad t \geq 0, \tag{30}$$

*where*

$$a_1 = \frac{-\alpha\delta}{2(1+\delta)}, \quad a_2 = \frac{\alpha}{1+\delta} + \frac{\beta(1+\delta)}{\alpha}. \tag{31}$$

*The initial and boundary conditions are taken from the exact solution (30).*

Numerical results of the LMM using the RK4 for Test Problem 5 is reported in Table 7. To verify the accuracy of the LMM, we have compared the results with the global meshless collocation method based on RBFs [15] and Adomian decomposition method [14]. The absolute errors for different $t$, $x$ and $N = 41$, $\Delta t = 0.001$, $\alpha = \beta = 0.001$, $\delta = 1$, spatial domain $[-20, 20]$ using IMQ RBF are given in Table 7. From the table, one can ensure that the results of the LMM are more accurate than the methods given in [14,15].

**Table 7.** Comparisons between the results obtained by the LMM with those adapted from [14,15] in term of $L_{abs}$ error norm for Test Problem 5.

| $t$ | $x$ | RK4 | [15] | [14] |
|---|---|---|---|---|
| | 0.1 | $3.2492 \times 10^{-8}$ | $2.7 \times 10^{-7}$ | $9.75 \times 10^{-6}$ |
| 0.005 | 0.5 | $3.2495 \times 10^{-8}$ | $1.4 \times 10^{-7}$ | $5.96 \times 10^{-5}$ |
| | 0.9 | $3.2498 \times 10^{-8}$ | $2.7 \times 10^{-7}$ | $9.75 \times 10^{-6}$ |
| | 0.1 | $2.4835 \times 10^{-9}$ | $2.7 \times 10^{-7}$ | $1.90 \times 10^{-5}$ |
| 0.01 | 0.5 | $2.4895 \times 10^{-9}$ | $1.3 \times 10^{-7}$ | $1.90 \times 10^{-5}$ |
| | 0.9 | $2.4955 \times 10^{-9}$ | $2.7 \times 10^{-7}$ | $1.90 \times 10^{-5}$ |

**Problem 6.** *The exact solution [27] of the 1D nonlinear standard Fitzhugh-Nagumo equation* (6) *is*

$$U(x,t) = \frac{1}{2} + \frac{1}{2} \tanh\left(\frac{1}{2\sqrt{2}}(x - \frac{2\rho - 1}{\sqrt{2}}t)\right), \quad x \in [-10, 10] \ t \geq 0. \tag{32}$$

*The initial and boundary conditions are taken from the exact solution* (32).

In Table 8 we have calculated numerical results for Test Problem 6 with $N = 101$, $\Delta t = 0.0001$ using MQ RBF. Table 8 shows $L_{rms}$ and $L_\infty$ error norms of the EEM for $q = 0.75$. From the table it can be seen that the obtained results are quite agreed with the results given in [27]. Figure 1 shows the comparison of numerical and analytical solutions for $t = 0.2, 0.4, 0.6, 0.8, 1$, $q = 4$ and $N = 41$ for Test Problem 6 while Figure 2 shows the numerical simulations of the EEM for $q = 0.75$ and $q = 4$.

**Table 8.** Comparisons between the results obtained by the LMM with those adapted from [27] for Test Problem 6.

| $t$ | $L_\infty$ EEM | $L_{rms}$ EEM | $L_\infty$ [27] | $L_{rms}$ [27] |
|---|---|---|---|---|
| 0.2 | $1.8896 \times 10^{-5}$ | $2.1960 \times 10^{-7}$ | $4.7416 \times 10^{-5}$ | $1.5880 \times 10^{-5}$ |
| 0.5 | $4.1554 \times 10^{-5}$ | $1.5696 \times 10^{-6}$ | $1.2312 \times 10^{-4}$ | $3.8433 \times 10^{-5}$ |
| 1.0 | $6.9891 \times 10^{-5}$ | $7.1449 \times 10^{-6}$ | $2.6261 \times 10^{-4}$ | $8.1870 \times 10^{-5}$ |
| 1.5 | $9.1687 \times 10^{-5}$ | $1.7262 \times 10^{-5}$ | $4.2096 \times 10^{-4}$ | $1.3387 \times 10^{-4}$ |
| 2.0 | $1.0969 \times 10^{-4}$ | $3.1857 \times 10^{-5}$ | $5.9999 \times 10^{-4}$ | $1.9433 \times 10^{-4}$ |
| 3.0 | $1.3942 \times 10^{-4}$ | $7.2878 \times 10^{-5}$ | $1.0324 \times 10^{-3}$ | $3.4320 \times 10^{-4}$ |
| 5.0 | $1.8964 \times 10^{-4}$ | $1.8803 \times 10^{-4}$ | $2.3050 \times 10^{-3}$ | $7.8638 \times 10^{-4}$ |

**Problem 7.** *The exact solution [29] of 1D coupled KdV equation* (7) *with* $\gamma = 3$ *and* $\alpha = \beta$ *is*

$$U(x,t) = \frac{\lambda}{\alpha} sech^2\left(\frac{1}{2}\sqrt{\frac{\lambda}{\alpha}}(x - \lambda t)\right), \quad V(x,t) = \frac{\sqrt{\frac{1}{\alpha}}\lambda sech^2\left(\frac{1}{2}\sqrt{\frac{\lambda}{\alpha}}(x - \lambda t)\right)}{\sqrt{2}}. \tag{33}$$

*The initial and boundary conditions are drawn out from the exact solution* (33).

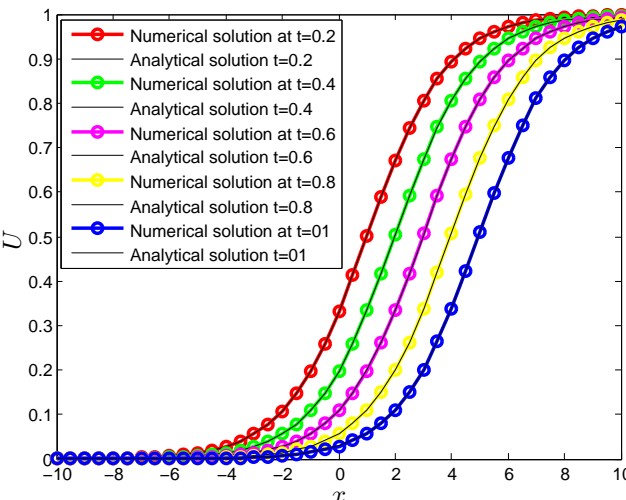

**Figure 1.** Comparing the curves of the numerical and analytical solutions for $N = 41$, $q = 4$ for Test Problem 6.

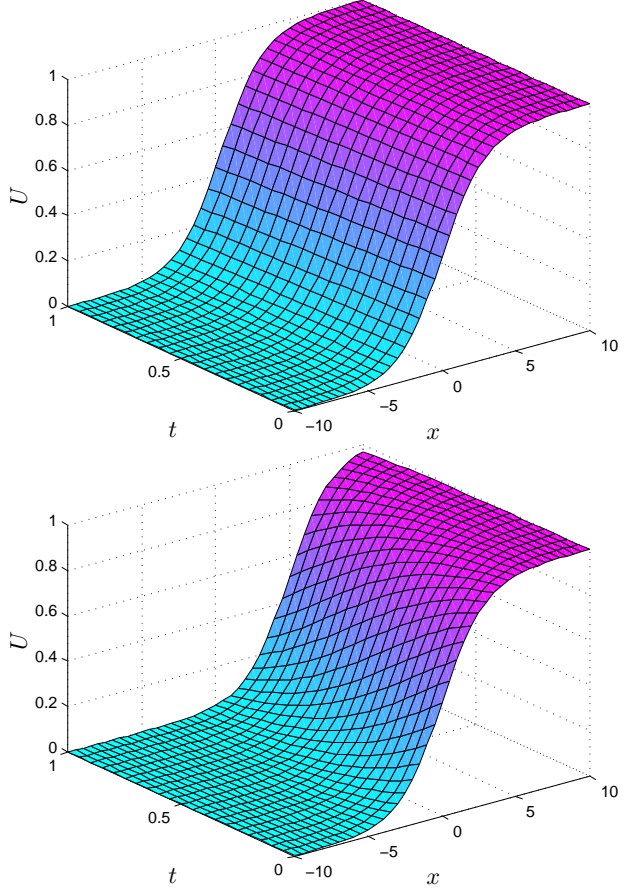

**Figure 2.** (**a**) Numerical solutions for $q = 0.75$; (**b**) Numerical solutions for $q = 4$ using the EEM for $N = 41$, $t = 1$ for Test Problem 6.

In Table 9, we have listed numerical simulations of the EEM versus results obtained from RBFs based collocation method [29] for Test Problem 7. The value of the parameters are $\alpha = \beta = \lambda = 0.01$ with the spatial domain $[-5, 5]$, $N = 101$, $\Delta t = 0.001$ and for various time $t$, using MQ radial basis function with $c = 100$. From the listed results given in Table 9, we have observed that the results obtained by the EEM are better than the results given in [29].

**Table 9.** Comparisons between the results obtained by the LMM with those adapted from [29] in term of $L_2$ error norm for Test Problem 7.

| t | 0.1 | 0.2 | 0.3 | 0.4 | 0.5 | 1 | 2 |
|---|---|---|---|---|---|---|---|
| [29] | | | | | | | |
| U | $3.662 \times 10^{-5}$ | $5.177 \times 10^{-5}$ | $4.065 \times 10^{-5}$ | $3.951 \times 10^{-5}$ | $3.964 \times 10^{-5}$ | $3.975 \times 10^{-5}$ | $4.461 \times 10^{-5}$ |
| V | $2.584 \times 10^{-6}$ | $3.648 \times 10^{-6}$ | $2.855 \times 10^{-6}$ | $2.769 \times 10^{-6}$ | $2.772 \times 10^{-6}$ | $2.749 \times 10^{-6}$ | $3.033 \times 10^{-6}$ |
| EEM | | | | | | | |
| U | $8.3147 \times 10^{-8}$ | $3.3251 \times 10^{-7}$ | $7.4809 \times 10^{-7}$ | $1.3299 \times 10^{-6}$ | $2.0779 \times 10^{-6}$ | $8.3112 \times 10^{-6}$ | $3.3245 \times 10^{-5}$ |
| V | $5.8794 \times 10^{-9}$ | $2.3512 \times 10^{-8}$ | $5.2898 \times 10^{-8}$ | $9.4037 \times 10^{-8}$ | $1.4693 \times 10^{-7}$ | $5.8769 \times 10^{-7}$ | $2.3508 \times 10^{-6}$ |

**Problem 8.** *The 1D Hirota-Satsuma coupled KdV system* (8) *with exact solution* [32] *given below*

$$U(x,t) = 4k^2 q_2 \tanh^2(\xi) - \frac{8k^2 q_2}{3} - \frac{C}{3},$$

$$V(x,t) = 2k^2 q_2 \tanh^2(\xi) - \frac{2k^2 q_2}{3} - \frac{4}{3} - c_0, \qquad (34)$$

$$W(x,t) = 2k^2 q_2 \tanh^2(\xi) - 2k^2 q_2 + c_0.$$

*where $\xi = \sqrt{q_2} k(x - Ct)$ and $a \le x \le b$.*
*The boundary and the initial conditions are obtained from the exact solutions* (34).

In Table 10 the numerical simulations of Hirota-Satsuma coupled KdV system (8) are carried out for Test problem 8 on the interval $[-30, 30]$ and different values of $k$ and $t$, with $N = 13$, $\Delta t = 0.05$, using MQ RBF with $c = 2$. The value of parameters $C = c_0 = q_2 = 0.1$. A full agreement between numeric and exact solution have been observed.

**Table 10.** Numerical results of the EEM in form of $L_\infty$ error norm for Test Problem 8.

| k | t | U | V | W |
|---|---|---|---|---|
| 0.1 | 0.25 | $4.9097 \times 10^{-5}$ | $1.7702 \times 10^{-8}$ | $1.7702 \times 10^{-8}$ |
| | 0.50 | $9.8195 \times 10^{-5}$ | $3.6410 \times 10^{-8}$ | $3.6410 \times 10^{-8}$ |
| | 0.75 | $1.4729 \times 10^{-4}$ | $5.6117 \times 10^{-8}$ | $5.6117 \times 10^{-8}$ |
| | 1.00 | $1.9639 \times 10^{-4}$ | $7.6814 \times 10^{-8}$ | $7.6814 \times 10^{-8}$ |
| | 5.00 | $9.8258 \times 10^{-4}$ | $6.0675 \times 10^{-7}$ | $6.0675 \times 10^{-7}$ |
| 0.05 | 0.25 | $5.2245 \times 10^{-6}$ | $2.2740 \times 10^{-9}$ | $2.2740 \times 10^{-9}$ |
| | 0.50 | $1.0464 \times 10^{-5}$ | $4.5679 \times 10^{-9}$ | $4.5679 \times 10^{-9}$ |
| | 0.75 | $1.5720 \times 10^{-5}$ | $6.8820 \times 10^{-9}$ | $6.8820 \times 10^{-9}$ |
| | 1.00 | $2.0991 \times 10^{-5}$ | $9.2162 \times 10^{-9}$ | $9.2162 \times 10^{-9}$ |
| | 5.00 | $1.0741 \times 10^{-4}$ | $4.9335 \times 10^{-8}$ | $4.9335 \times 10^{-8}$ |

**Problem 9.** *The analytic solution of the 2D reaction-diffusion Brusselator system* (9) *for a particular case in the region* $(x, y) \in [0, 1]^2$, $t \ge 0$ *with* $\alpha = 1$, $\beta = 0$, $\gamma = 0.25$ *is given in* [41]

$$U(x, y, t) = \exp\left(-x - y - \frac{t}{2}\right), \quad V(x, y, t) = \exp\left(x + y + \frac{t}{2}\right). \qquad (35)$$

*The initial and boundary condition are taken from the exact solution* (35).

The LMM is employed for the numerical solution of Test Problem 9 by letting time step length $\Delta t = 0.001$, the shape parameter value $c = 1$, $N = 20 \times 20$, at various times up to $t = 1.8$. In Table 11 we have compared the results obtained by the EEM with the exact solution as well as with [38,41]. Reasonably good accuracy has been obtained in this case as well also CPU time in seconds are reported in the same table.

The numerical results on irregular domains are shown in Figures 3–6 for Test Problem 9 using MQ RBF with shape parameter $c = 1$. The numerical solutions shown in Figures 3–6 are performed

with $\Delta t = 0.001$, $t = 1$ $\alpha = 1$, $\beta = 0$ and $\mu = 0.25$. These figures show the efficiency of the suggested method in irregular geometry in term of absolute error $L_{abs}$ by using the EEM for Test Problem 9.

**Table 11.** Comparisons between the results obtained by the LMM with those adapted from [38,41] using GA RBF at point $(0.40, 0.60)$ for Test Problem 9.

| t | CPU Time | U | | | | V | | | |
|---|---|---|---|---|---|---|---|---|---|
| | | EEM | [38] | [41] | Exact | EEM | [38] | [41] | Exact |
| 0.30 | 0.17 | 0.3167 | 0.3168 | 0.3166 | 0.3166 | 3.1584 | 3.158 | 3.157 | 3.158 |
| 0.60 | 0.19 | 0.2726 | 0.2724 | 0.2725 | 0.2725 | 3.6696 | 3.669 | 3.667 | 3.669 |
| 0.90 | 0.23 | 0.2346 | 0.2347 | 0.2345 | 0.2346 | 4.2635 | 4.263 | 4.260 | 4.263 |
| 1.20 | 0.26 | 0.2019 | 0.2020 | 0.2018 | 0.2019 | 4.9534 | 4.953 | 4.950 | 4.953 |
| 1.50 | 0.30 | 0.1738 | 0.1739 | 0.1737 | 0.1738 | 5.7551 | 5.755 | 5.751 | 5.755 |
| 1.80 | 0.34 | 0.1496 | 0.1496 | 0.1495 | 0.1496 | 6.6864 | 6.686 | 6.681 | 6.686 |

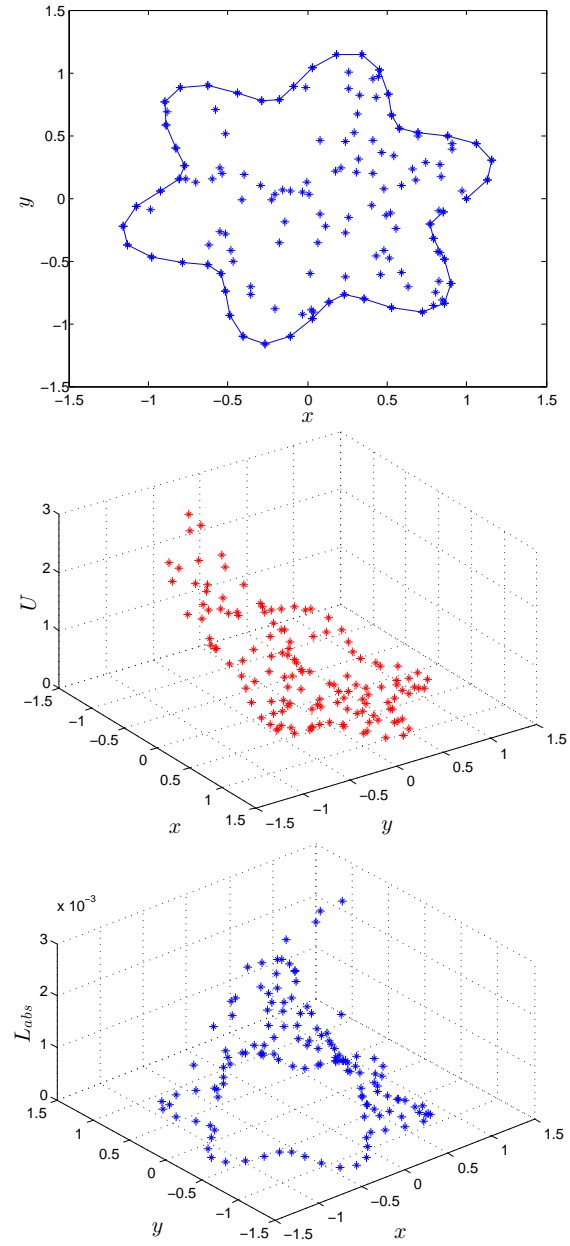

**Figure 3.** Computational domain (**a**), numerical solution (**b**) and absolute error (**c**) by using the EEM for Test Problem 9.

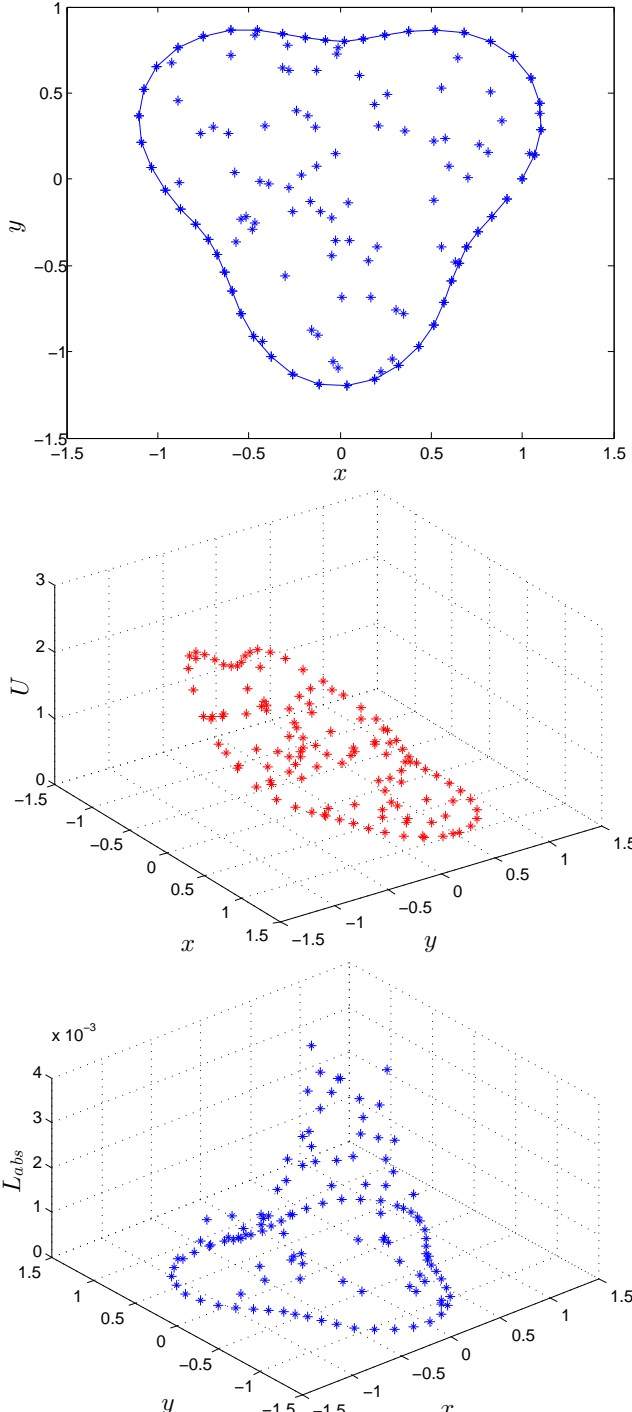

**Figure 4.** Computational domain (**a**), numerical solution (**b**) and absolute error (**c**) by using the EEM for Test Problem 9.

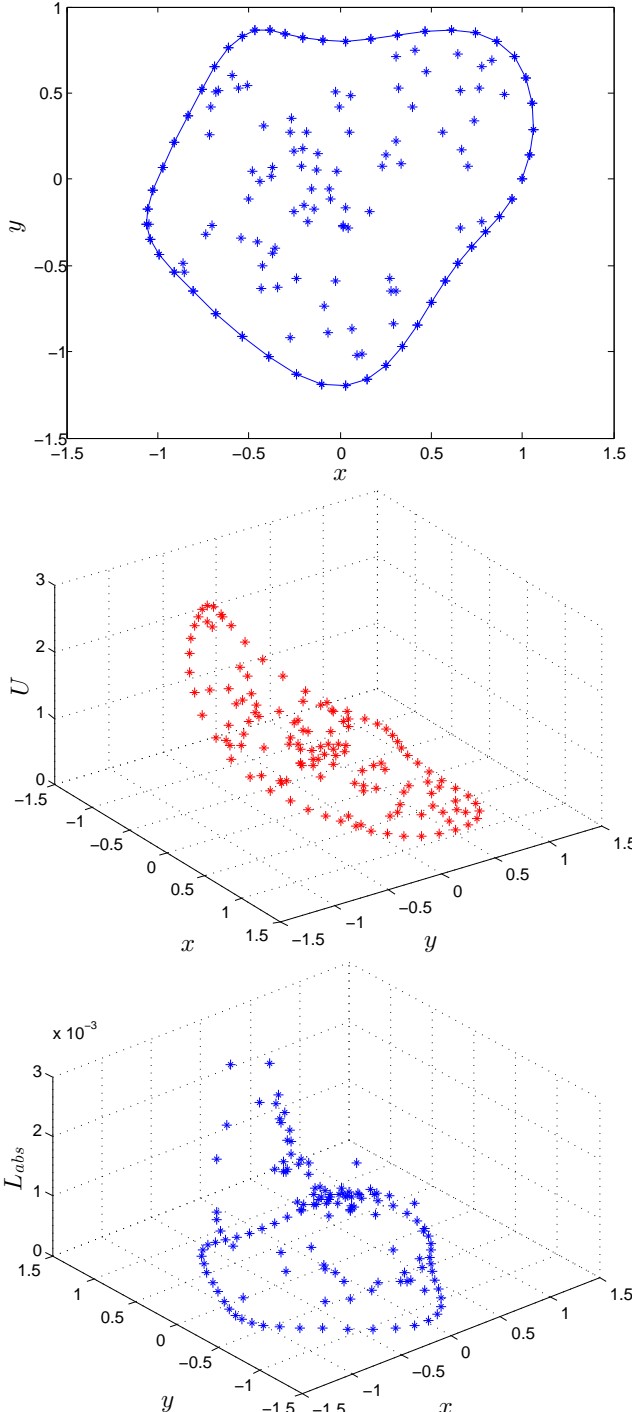

**Figure 5.** Computational domain (**a**), numerical solution (**b**) and absolute error (**c**) by using the EEM for Test Problem 9.

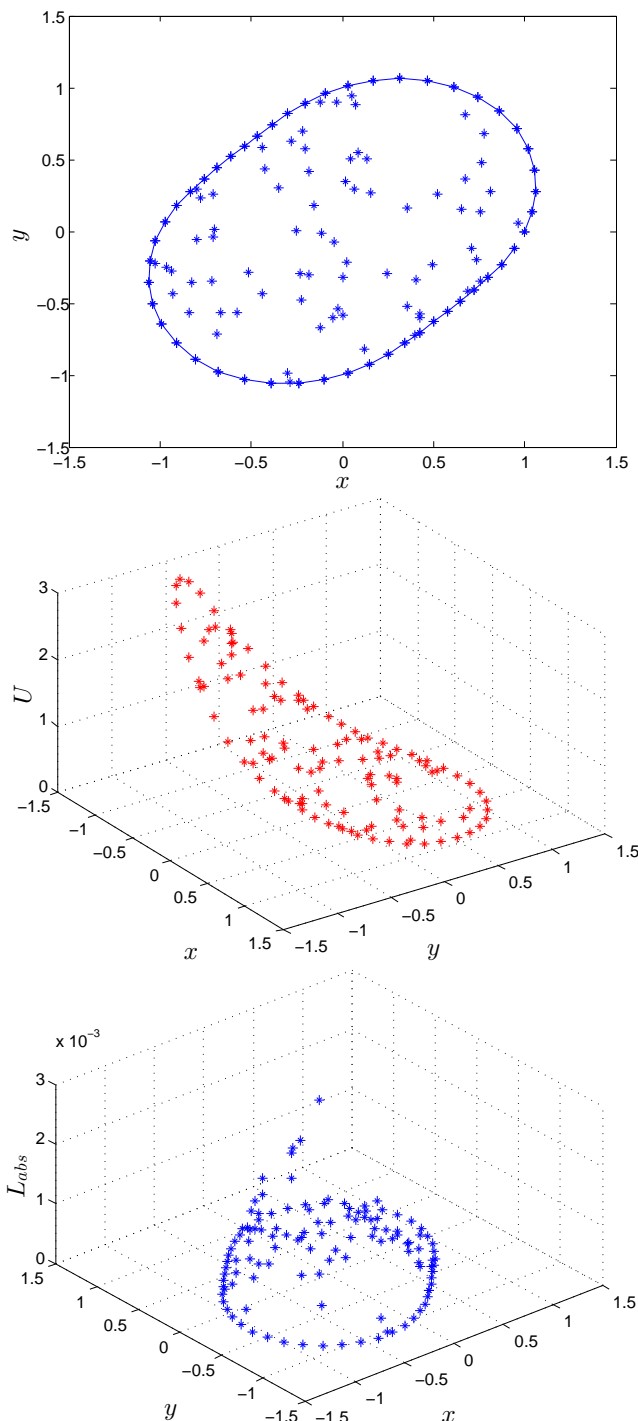

**Figure 6.** Computational domain (**a**), numerical solution (**b**) and absolute error (**c**) by using the EEM for Test Problem 9.

## 5. Conclusions

In this work, a local meshless differential quadrature collocation method is proposed for numerical solution of different mathematical models arising in science and engineering. These models have been solved in the literature by using various numerical methods but our proposed scheme LMM is efficient and accurate on both regular and irregular domains. Results of the LMM are compared with exact or approximate solutions (which ever is available) in the existence literature. On the basis of these results and comparison, we can conclude that the proposed local meshless method is accurate, efficient and

its implementation is very simple, straightforward, irrespective of the dimension and geometry of the problem in hand.

**Author Contributions:** Conceptualization, I.A and M. Ahsan.; Methodology, I.A.; Software, I.A.; Validation, I.A., M.A. (Masood Ahmad), Z.-u.D. and M.A. (Muhammad Ahsan); Formal Analysis, P.K. and Z.-u.D.; Investigation, P.K., I.A., M.A. (Muhammad Ahsan) and M.A. (Masood Ahmad); Resources, P.K.; Data Curation, I.A.; Writing—Original Draft Preparation, I.A.; Writing—Review & Editing, I. A., P.K. and Z.-u.D.; Visualization, I.A.; Supervision, I.A. and P.K; Project Administration, P.K.; Funding Acquisition, P.K.

**Funding:** This project was supported by the Theoretical and Computational Science (TaCS) Center under Computational and Applied Science for Smart Innovation Cluster (CLASSIC), Faculty of Science, KMUTT.

**Acknowledgments:** The authors thank Editors and Referees for their valuable comments and suggestions, which were very useful to improve the paper significantly. The first author would like to thank the Theoretical and Computational Science (TaCS) Center under Computational and Applied Science for Smart Innovation research Cluster (CLASSIC), Faculty of Science, KMUTT. Moreover, the authors acknowledge the financial support provided by King Mongkut's University of Technology Thonburi, Thailand through the "KMUTT 55$^{th}$ Anniversary Commemorative Fund".

**Conflicts of Interest:** The authors declare no conflict of interest.

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
