# Peer review of "An Efficient Local Formulation for Time–Dependent PDEs"

_mathematics, doi:10.3390/math7030216_

Round 1

Reviewer 1 Report

The presentation of the article really needs to get improved not only in terms of English but also in terms of style. The sentences are not connected, which makes the article hard to read. The Results are worth being published but only after re-writing the paper. Also, I have some concern about the novelty of the work. How do the authors compare the present work with previous ones on this topic? What is the main goal and achievement of this paper which makes it different from them?

Author Response

Reviewer # 1:

The presentation of the article really needs to get improved not only in terms of English but also in terms of style. The sentences are not connected, which makes the article hard to read. The Results are worth being published but only after re­writing the paper. Also, I have some concern about the novelty of the work. How do the authors compare the present work with previous ones on this topic? What is the main goal and achievement of this paper which makes it different from them?

Our Response:            We have incorporated the suggestions of  the honorable referee.  We have corrected some typographical mistakes and a language expert is consulted. Abstract, Introduction, conclusion of the paper has been modified also to try our best to present the numerical method and results in more clearer manner.  

This manuscript concerning one and two dimensional time-dependent PDEs. Accuracy, efficacy the proposed LMM are  shown  by several  problems in regular and irregular domains using uniform and non uniform nodes. The numerical results for different problems have been compared with almost 10 different methods  and our results are founded superior.  The paper will be useful for a lot of scientists working in new trends of meshless methods. The present work will lay a foundation for numerical solution of PDEs that have been got broadly attention in almost all disciplines of engineering and, then, possesses biological implications and extends the former researches.

Unlike Traditional numerical methods, such as finite element, finite difference, or finite volume methods,  Meshless methods  uses radial distance r to realize numerical solution of the problem. This is achieved by composing some univariate basic function with a (Euclidean) norm, and therefore turning a problem involving many space dimensions into one that is virtually one-dimensional.

So in the case of meshless methods going from one –dimensional case to higher dimensional (2-, 3-, 4- dimensions etc.) is not a big deal. The method structure remainsthe same. Only the size of the matrix increases. The only change which is needed is the change in the radial function.

In one dimensional case, the radial function is defined as   . In three- dimensional case it becomes  and so on. Hence, irrespective of the dimensionality of the problem, we are dealing with one variable r. Of course, the size of the matrix increases in higher dimensions but the structure the method is preserved. This is one of the advantage of meshless method over the other traditional numerical methods.

Reviewer 2 Report

In the paper, a local meshless method (LMM) based on radial basis functions (RBFs) is utilized for the numerical solution of various types of PDEs, due to the flexibility with respect to geometry and high order of convergence rate.

In case of global meshless methods, the two major deficiencies are the computational cost and the optimum value of shape parameter. Therefore, research is currently focus towards localized radial basis function approximations, as the local meshless method is proposed here. The local meshless procedures is used for spatial discretization whereas for temporal discretization different time integrators are employed. The proposed local meshless method is testified in terms of efficiency, accuracy and ease of implementation using regular and irregular domains.

In my idea the English of the paper needs to be polishing.

Author Response

Reviewer # 2

The authors demonstrate the efficiency of a local meshless method based on radial basis functions for numerical solution of various types of PDEs. The authors consider equations appeared mainly in hydrodynamics. They note the flexibility of the method with respect to geometry and high order of convergence rate. The local meshless method is testified in terms of efficiency, accuracy and ease of implementation using regular and irregular domains for many examples. meshless method is testified in terms of efficiency, accuracy and ease of implementation using regular and irregular domains.

Reviewer Comment:              In my idea the English of the paper needs to be polishing .
Our Response:            We have incorporated the suggestion and a language expert is consulted.

Reviewer 3 Report

The authors demonstrate the efficiency of a local meshless method based on radial basis functions for numerical solution of various types of PDEs. The authors consider equations appeared mainly in hydrodynamics. They note the flexibility of the method with respect to geometry and high order of convergence rate. The local meshless method is testified in terms of efficiency, accuracy and ease of implementation using regular and irregular domains for many examples.

 In my opinion, the article is clearly written, it is interesting to the readers and contain new results. It can be accepted in the present form.

Author Response

Reviewer # 3

The authors demonstrate the efficiency of a local meshless method based on radial basis functions for numerical solution of various types of PDEs. The
authors consider equations appeared mainly in hydrodynamics. They note the flexibility of the method with respect to geometry and high order of convergence rate. The local meshless method is testified in terms of efficiency, accuracy and ease of implementation using regular and irregular domains for many examples.

Reviewer Comment:              In my opinion, the article is clearly written, it is interesting to the readers and contain new results. It can be accepted in the present form.

Our Response:            We wish to acknowledge the anonymous referees whose suggestions have helped improve the quality of our paper.
